# Upgrading the Properties of Reduced Graphene Oxide and Nitrogen-Doped Reduced Graphene Oxide Produced by Thermal Reduction toward Efficient ORR Electrocatalysts

**DOI:** 10.3390/nano9121761

**Published:** 2019-12-11

**Authors:** Carolina S. Ramirez-Barria, Diana M. Fernandes, Cristina Freire, Elvira Villaro-Abalos, Antonio Guerrero-Ruiz, Inmaculada Rodríguez-Ramos

**Affiliations:** 1Instituto de Catálisis y Petroleoquímica, CSIC, Marie Curie 2, 28049 Madrid, Spain; ramirez.carolina@outlook.com; 2Dpto. Química Inorgánica y Técnica, Facultad de Ciencias UNED, Senda del Rey 9, 28040 Madrid, Spain; 3REQUIMTE/LAQV, Departamento de Química e Bioquímica, Faculdade de Ciências, Universidade do Porto, 4169-007 Porto, Portugal; acfreire@fc.up.pt; 4Interquimica, Antonio de Nebrija 8, 26006 Logroño, Spain; evillaro@interquimica.org

**Keywords:** graphite, reduced graphene oxide, nitrogen-doped reduced graphene oxide, exfoliation, oxygen reduction reaction, electrocatalysis

## Abstract

N-doped (NrGO) and non-doped (rGO) graphenic materials are prepared by oxidation and further thermal treatment under ammonia and inert atmospheres, respectively, of natural graphites of different particle sizes. An extensive characterization of graphene materials points out that the physical properties of synthesized materials, as well as the nitrogen species introduced, depend on the particle size of the starting graphite, the reduction atmospheres, and the temperature conditions used during the exfoliation treatment. These findings indicate that it is possible to tailor properties of non-doped and N-doped reduced graphene oxide, such as the number of layers, surface area, and nitrogen content, by using a simple strategy based on selecting adequate graphite sizes and convenient experimental conditions during thermal exfoliation. Additionally, the graphenic materials are successfully applied as electrocatalysts for the demanding oxygen reduction reaction (ORR). Nitrogen doping together with the starting graphite of smaller particle size (NrGO_325_-4) resulted in a more efficient ORR electrocatalyst with more positive onset potentials (*E*_onset_ = 0.82 V versus RHE), superior diffusion-limiting current density (*j*_L, 0.26V, 1600rpm_ = −4.05 mA cm^−2^), and selectivity to the direct four-electron pathway. Moreover, all NrGO_m_-4 show high tolerance to methanol poisoning in comparison with the state-of-the-art ORR electrocatalyst Pt/C and good stability.

## 1. Introduction

In recent years, the interest in graphene has grown due to its many outstanding electronic, thermal, chemical, and mechanical properties [1]. These properties give graphene an enormous versatility. It can be extensively used in diverse applications including energy conversion and storage technologies [2,3,4,5], electronics [6], and sensing device applications [7]. Graphenic materials have also been widely employed as solid (electro)catalytic materials, either as active phases or as supports [8]. These (electro)catalytic applications strongly depend on its surface chemical properties. It is well known that the surface chemical properties of graphenic materials can be tuned by covalent added adatoms [9]. Thus, the presence of nitrogen or boron atoms in the basal planes of graphene layers produce changes to many of their chemical properties. It was proven that the doping of nitrogen into a graphenic nanostructure modifies its chemical and electrical properties, [10]. Thus, the N atoms inserted in the graphene structure increase the electronic density of the graphenic layer, since one more electron is added to the graphenic layer, making it more basic [11,12]. In addition, the easily tunable surface chemical properties and specific surface area of the graphenic materials make them very versatile materials for application in this field.

An extensive number of works have been devoted to the preparation of graphene and N-doped graphene. The synthesis of graphene can be performed using different methods, among them the micromechanical cleavage method [13], chemical vapor deposition (CVD) [14], and epitaxial growth on silicon carbide surfaces [15]. However, they show low productivity and lack of properties’ selectivity. One of the common approaches used for a large-scale graphene production is based on the chemical oxidation of graphite (G) flakes to produce graphite oxide (GO) [16,17] with strong oxidizing agents. GO can be later exfoliated and converted to reduced graphene oxide (rGO) through a reduction procedure. The most used routes for the reduction of GO are chemical [18,19], electrochemical [20,21], solvothermal [22,23], and thermal treatments [24]. The use of thermal reduction is preferred over the rest of the methods due to its simplicity and industrial scalability [25].

The incorporation of nitrogen in graphenic materials to produce N-doped reduced graphene oxide (NrGO) has been described using several methods as well. For example, CVD using NH_3_ [26], acetonitrile [27], or pyridine [28] as the N source; the arc discharge of graphite [29] in the presence of pyridine or NH_3_; the nitrogen plasma treatment of graphene [30]; and the thermal treatment of GO with melamine [31,32], urea [33], or NH_3_ [9,34].

The improvement of some of the properties of rGO and NrGO has been undertaken using different synthetic strategies. Wu et al. [35] studied a chemical exfoliation of GO to produce reduced graphene oxide with a selective number of layers based on different starting materials such as pyrolytic graphite, natural flake graphite, Kish graphite, flake graphite powder, and artificial graphite. They reported the effect of the lateral size and crystallinity of these starting graphite materials on the number of graphene layers presented in the obtained graphenic material. They found that graphite samples with a small lateral size and low crystallinity produce a higher proportion of single layer graphene. Li et al. [34] obtained NrGO through the thermal annealing of GO in NH_3_ sweeping temperatures between 300 and 1100 °C. Annealing at 500 °C afforded the highest N-doping level of ∼5%, showing the strong influence of the temperature on the nitrogen content of the obtained materials. They claim that the N-doping degree depends on the amount of oxygen functional groups of graphene as they are responsible for the formation of C−N bonds. The higher the annealing temperature is, the lower the content of oxygen, leading to a lower reactivity between the graphene layer and NH_3_. The production of NrGO based on the annealing of GO in the presence of melamine at high temperature (700−1000 °C) has been reported by Sheng et al. [32]. They noted that nitrogen content depends on the mass ratio between GO and melamine as well as on the temperature, reaching values of 10.1 at % using a 1:5 ratio of GO to melamine at 700 °C. A similar approach was used by Canty et al. [33] working with urea and GO as precursor materials of NrGO. They used the ratio of GO to urea as a way to control the amount of nitrogen inserted and the surface area values obtained. Nonetheless, neither Li et al. [34] nor Canty et al. [33] reported a systematic study of the synthesis conditions to optimize NrGO properties. Menendez’s group [25] analyzed the effect of the experimental conditions of the thermal transformation of GO onto rGO, finding that the treatment temperature strongly affects the type and amount of functional groups obtained. Following this line of research, they proposed that the temperature of the initial flash thermal treatment allows the control of the surface area obtained [36]. However, the surface area values achieved were lower than 500 m^2^g^−1^. Zhang et al. [37] proposed a vacuum-promoted thermal exfoliation method for different samples of GO, which were obtained from natural flake graphites with particle sizes ranging from 100 to 5000 mesh. Nevertheless, neither the morphology nor the structures of different graphenic samples were affected by the parent graphite particle size. The surface areas observed were around 490 m^2^g^−1^ and the C/O ratio determined by XPS analysis revealed that all the samples had almost the same oxygen content. Thus, the vacuum-promoted exfoliation method minimizes the differences originated from the raw graphite particle size. The effect of the raw graphite size on the rGO properties has been also described by Dao et al. [38]. They prepared rGO by the rapid heating of dry GO using three graphite particle sizes obtained from grinding a large graphite sample. An increase in the surface area was observed as the particle size of the samples was reduced, reaching a value of 739 m^2^g^−1^ in their best sample. This improvement was explained as being due to the better oxidation degree achieved with the decrease of the graphite particle size, which also favors a better exfoliation of the GO.

On the other hand, a previous study of our group [10] reported the synthesis of NrGO from GO obtained from three different graphite particle sizes. We found that the quantity of nitrogen and the surface properties obtained were dependent on the particle size of the graphite used. However, the surface areas obtained were not optimized enough.

Due to the lack of simultaneous and systematic reports regarding the combined effect of raw graphite size and the conditions of the thermal treatment (heating rate and temperature) on the properties of non-doped and N-doped reduced graphene oxide, we have investigated the preparation of graphenic materials using three different particle sizes of starting graphite and using various thermal treatments. Moreover, a deep characterization of the graphenic materials was performed before applying those as electrocatalysts (ECs) for the oxygen reduction reaction, in order to optimize the properties of the obtained materials—particularly, the surface area and content of nitrogen—due to the importance of these parameters in their electrocatalytic performance. Finally, the synthesized materials have been tested as ECs in the oxygen reduction reaction (ORR).

## 2. Materials and Methods 

### 2.1. Preparation and Characterization of Graphenic Materials

Graphenic materials were prepared by the treatment of graphite oxide (GO) at high temperature. Natural graphite powders of various grain sizes—10 mesh, 100 mesh, and 325 mesh (supplied by Alfa Aesar, Thermo Fisher Scientific, UK, purity 99.8%) were used for the synthesis of GO through a modified Brodie’s method [17]. This procedure consists of the addition of 10 g of graphite (G) over 200 mL of fuming HNO_3_, keeping the mixture at 0 °C. First, 80 g of KClO_3_ was gradually added over 2 h. Afterward, the mixture was stirred for 21 h, maintaining the 0 °C temperature. The GO obtained was filtered and washed systematically with water until neutral pH and dried under vacuum at room temperature. The resultant samples were labeled GO_m_, where m indicates the mesh size used.

The GO_m_ were exfoliated in a vertical quartz reactor under two different atmospheres. The first atmosphere consists of nitrogen (87 mL/min) producing rGO, while the second was a mixture of NH_3_, H_2_, and N_2_ (10, 3, and 87 mL/min, respectively) generating NrGO.

In order to study the effect of temperature and heating rate on the properties of rGO and NrGO, for each atmosphere described, five different exfoliation ramps were applied over GO_325_. In a first ramp, 0.3 g of GO was introduced in the furnace and heated at 5 °C min^−1^ to 250 °C; then, the sample was kept at this temperature for 30 min. The temperature was increased from 250 to 500 °C with a heating rate of 5 °C min^−1^ and then kept at this temperature for 30 minutes. In a second and third ramp, GO was heated at 5 °C min^−1^ and 10 °C/min respectively to 250 °C; then, the sample was kept at this temperature for 30 min. The temperature was increased from 250 to 700 °C using the same heating rates, and then this temperature was maintained for 30 min. In a fourth ramp, GO was heated at 5 °C min^−1^ to 100 °C; then, the sample was kept at this temperature for 1 h. The temperature was increased from 100 to 700 °C with a heating rate of 10 °C min^−1^; then, the sample was kept at this temperature for 5 min. Finally, in the fifth ramp, the GO was heated up to 250 °C at 20 °C min^−1^; then, the sample was kept at this temperature for 30 min. The temperature was subsequently increased from 250 to 500 °C with a heating rate of 20 °C min^−1^; then, the sample was kept at this temperature for 30 min.

For the exfoliation of the samples GO_10_ and GO_100_, the temperature and heating rate used were selected according to the ramp that gave the higher value of surface area among the exfoliated samples of GO_325_. The samples obtained were labeled rGO_m_-r and NrGO_m_-r, respectively, where m indicates the mesh size used and r indicates the ramp used.

The characterization of the as-prepared materials was carried out by different techniques: elemental analysis, nitrogen adsorption isotherms, X-ray diffraction (XRD), scanning electron microscopy (SEM), transmission electron microscopy (TEM), and X-ray photoelectron spectroscopy (XPS). The detailed equipment and methods used are described in the Appendix A.

### 2.2. ORR Electrocatalytic Activities

For the cyclic voltammetry (CV) and linear sweep voltammetry (LSV) experiments, an Autolab PGSTAT 302N potentiostat/galvanostat (EcoChimie B.V. Netherlands) was used. A modified glassy carbon rotating disk electrode, RDE (*d* = 3 mm, Metrohm, Switzerland) was used as the working electrode, a carbon rod (*d* = 2 mm, Metrohm, Switzerland) was used as the counter, and an Ag/AgCl (3 mol dm^−3^ KCl, Metrohm, Switzerland) was used as the reference electrode.

Before any type of modification, the electrode was cleaned (detailed information in ESI). The electrode modification was achieved by dropping 2 × 2.5 µL of electrocatalyst dispersion onto the RDE surface followed by solvent evaporation under a flux of hot air. To prepare the electrocatalyst dispersion, 1 mg of material was mixed with 20 μL of Nafion, 125 μL of ultrapure water, and 125 μL of isopropanol. Then, the mixture was sonicated for 15 min.

The CV and LSV tests were performed in KOH electrolyte (0.1 mol dm^−3^) saturated in O_2_ and N_2._ (30 min purge for each gas). The potential range used for CV and LSV tests was between 0.26 and 1.46 V versus reversible hydrogen electrode (RHE) at a scan rate of 0.005 V s^−1^. In addition, for the LSV tests, rotation speeds between 400 and 3000 rpm were used. The effective ORR current was determined through the subtraction of the current obtained in KOH saturated with N_2_ by that saturated with O_2_. The stability was evaluated by chronoamperometry (CA) for 20,000 s at 0.5 V versus RHE and 1600 rpm. CA was also used to determine the tolerance to methanol crossover by applying a *E* = 0.5 V versus RHE for 2000 s and a 1600 rpm rotation speed.

All relevant ORR parameters (effective currents, diffusion-limiting current densities (*j*_L_), onset potential (*E*_onset_), Tafel slope, and the number of electrons transferred for each O_2_ molecule (*ñ*_O2_)) were calculated as described in the ESI file. 

Experiments with the rotating ring disk electrode (RRDE) were also conducted in KOH electrolyte saturated with O_2_ to determine the amount of H_2_O_2_ produced. This was achieved using Equation (1) where *i*_R_ and *i*_D_ correspond to the ring and disk currents, respectively, and N is the current collection efficiency of the Pt ring (*N* = 0.25) [39].
(1)$${\%\;H}_{2}O_{2}\;=\;200\;\times \;\frac{i_{R}/\;N}{i_{D}+i_{R}/N},$$

## 3. Results and Discussion

### 3.1. Characteristics of the Graphenic Materials

XRD is a powerful tool to evaluate the interlayer changes of graphene-based materials. The XRD patterns of G_10_, G_100_, G_325_, GO_10_, GO_100_, and GO_325_ samples are shown in Figure 1. The G samples presented the characteristic diffraction peak corresponding to pristine graphite (002) reflection at 2θ~26° [10].

A downshift for (002) reflection peak to 2θ~15° after the oxidation treatment was observed for all the GO samples. It indicates that a successful oxidization of all G samples was achieved [8]. The distance between layers (d_(002)_) increased from 0.33 nm (for G) to 0.57 nm (for GO samples), which is due to the presence of interlayered species incorporated during the oxidation of graphite [25].

The morphology of the GOs was observed by SEM. Figure 2 shows representative images of the GO samples. The analysis of the particle size distribution based on a minimum of 200 particles shows that the average particle sizes of GO_10_, GO_100_, and GO_325_ samples are 86, 38, and 25 µm, respectively.

Slow heating rates were selected to be evaluated during the exfoliation process of GOs, since fast heating rates produce more wrinkled sheets [25]. Thus, small heating rates are fast enough to produce an effective expansion allowing the exfoliation and minimizing the distortion of the graphene sheets. It is known that oxygen groups decompose at high temperatures, reducing the number of reactive sites for N doping [34], and that high annealing temperatures (>700 °C) could break C–N bonds in the NrGO materials leading to a low doping level [40]. So, temperatures up to 700 °C were used in the thermal treatments applied to the GO samples.

Table 1 compiles the main characteristics of the non-doped and N-doped reduced graphite oxides obtained. Firstly, the GO_325_ sample was submitted in both inert atmosphere and atmosphere with the presence of ammonia to five different thermal ramps described in the experimental section. Application of the BET method to N_2_ adsorption isotherms (these type IV isotherms were displayed in the Appendix A of the Electronic Supplementary Information, ESI) measured over the GO_325_-derived non-doped and N-doped reduced graphene oxides gave surface area (S_BET_) values ranging from 667 to 867 m^2^ g^−1^ for the different rGO_325_ samples and from 427 to 492 m^2^ g^−1^ for the NrGO_325_ samples. These obtained S_BET_ values are significantly lower than the theoretical value calculated for a single layer of graphene (2630 m^2^ g^−1^) [8]. This finding indicates the piling up of graphene layers and the formation of a few-layer graphene structure for both rGO and NrGO. However, these values are much higher than the values previously reported using thermal exfoliation to produce rGO [36,38] and NrGO [10]. From the results shown in Table 1, it can be seen that ramp 3 for non-doped reduced graphene oxide (rGO_325_) and ramp 4 for N-doped reduced graphene oxide (NrGO_325_) led to reduced materials having an enhanced S_BET_. The ramps used for the exfoliation of GO_10_ and GO_100_ were selected based on these results. Thus, the ramp used for the preparation of rGO_10_ and rGO_100_ was ramp 3, and for NrGO_10_ and NrGO_100_ was ramp 4. A significant and gradual decrease of the surface area values was observed for rGO and NrGO sample series as the size of starting graphite increases. These findings are coherent with the tendency previously reported for non-doped graphene by Dao et al. [38] where the lower the starting G size, the higher the oxidation degree of the obtained GO, which could lead to a better exfoliation of rGO.

The completion of the exfoliation process for the prepared graphene materials was investigated by XRD. Examination of the rGO_325_-r and NrGO_325_-r patterns (Appendix A ESI) indicates that the selected exfoliation ramps, 3 (inert) and 4 (ammonia), conduce to a successful exfoliation of the GO325 sample since the characteristic diffraction peak at 2θ~15° of GO (see Figure 1) disappeared, suggesting a reduction of GO325 to NrGO325 and rGO325 samples [10]. In addition, the absence of reflections corresponding to crystalline graphite for GO_325_-3 and NrGO_325_-4 samples supports the adequacy of ramps 3 (inert) and 4 (ammonia). The greater or lesser success in the GO exfoliation process can be evaluated by the appearance or not of crystalline graphite reflections. Figure 3 shows the XRD patterns of NrGO_m_-4 and rGO_m_-3. It can be seen that the exfoliation process was fully successful in inert atmosphere for all the GO and also in ammonia atmosphere, especially for the GO_100_ and GO_325_ samples, because only a small and broad peak appeared at 2θ slightly lower than 26°, corresponding to the graphite (002) reflection. This fact indicates that the sample contains some small restacking of graphene layers and the formation of a few-layer graphene structures. The intensity of this peak displays a progressive increase with the particle size of the GO starting material, showing a higher number of restacked layers (rGO_10_ > rGO_100_ ≥ rGO_325_). Particularly, the progressive association of the graphene sheets produced after blasting is favored under ammonia reactive conditions but without reaching the level typical for crystalline graphite. This tendency is coherent with the values of S_BET_. The average stacking number of graphene layers (N_L_) in the exfoliated samples was estimated by using the layer-to-layer distance (d_(002)_) and the size of the crystallite measured from the width of the diffraction peak of the (002) reflection, using the Scherrer equation [12]. The N_L_ calculated show higher values than the ones expected according to the S_BET_ obtained. This may be due to the fact that the XRD signal is strongly influenced by the thicker particles, because although these constitute a minor proportion in the total of the sample, they are the ones with the highest diffraction signal. The X-ray crystalline parameters are shown in Table 1.

Some morphological evidences of the differences between rGO and NrGO were determined by TEM (Figure 2). The rGO TEM images exhibited the presence of winkled structures of graphene. The introduction of nitrogen in the graphitic structure—the NrGO samples—did not produce noticeable difference in the morphology of the graphene sheets. High resolution transmission electron microscopy (HRTEM) characterization further showed that rGO_m_-3 and NrGO_m_-4 samples consist of 5–12 graphene layers (Appendix A).

The elemental analysis technique was used to determine the amount of nitrogen atoms in the graphene materials (see Table 1 for N wt %). For the rGO sample, nitrogen was not found. The NrGO_325_-4 sample presents the higher N content among the NrGO samples with 5.0 wt %, and a gradual decrease of the N content values was observed as the size of starting graphite increases. There are two explanations for this phenomenon. First, this tendency can be explained in terms of the higher degree of oxidation obtained in GO samples with smaller graphite particle sizes [38]. It is known that the oxygen functional groups in GO including carbonyl, carboxylic, lactone, and quinone groups are responsible for reacting with NH_3_ to form C–N bonds, allowing the incorporation of N in the structure [34]. Thus, a smaller graphite particle size favors the formation of GO with a higher degree of oxidation, which could lead to higher N contents. Besides, the higher degree of exfoliation of small crystals also facilitates the contact of the ammonia with the oxygen groups in the basal plane of the sheets, leading to a better incorporation of N atoms.

XPS is a powerful technique to identify the chemical states of the surface species. It was used to analyze the surface of the different graphene materials from the characteristic XPS peaks corresponding to C, N, and O regions (Appendix A ESI shows general XPS spectra). The results obtained from the analysis of the C1s, O1s, and N1s individual high-resolution spectra are shown in Table 2. The assignment of the components of the N1s, O1s, and C1s region is not straightforward. The value of binding energy observed for the different functional groups of these elements varies in the literature. It may be due to the specific environments of the atoms and the redistribution of electrons after the ionization of the sample [30,41,42,43]. Nitrogen peak deconvolution for the NrGO_m_-4 samples (Figure 4) indicated the presence of four elementary peaks: pyridinic nitrogen (399.5–398.5 eV), pyrrolic nitrogen (400.8–399.8 eV), quaternary nitrogen (403.0–401.0 eV), and NOx groups (404.9–405.6 eV) [10,44,45,46,47,48]. XPS analysis indicated that about 3.2–3.4 at % N was found in the surface of the graphene sheets after ammonia treatment. These values are lower than those corresponding to the bulk N% content obtained by elemental analysis. This difference could be attributed to inhomogeneous nitrogen doping of the graphenic materials, since it was inferior at the surface analyzed by XPS. For rGO samples, the nitrogen peak was undetected. The percentage of pyridinic N species was slightly higher for the samples obtained from smaller sizes and quaternary nitrogen was higher for the samples obtained from bigger ones. The pyridinic nitrogen is the most basic among the different N species. Therefore, the basicity of carbon catalysts is related to the content of pyridinic groups [48,49].

The C1s spectra was solved considering five components, which can be assigned to graphitic sp^2^ carbon atoms (284.6–285.1 eV), the C–O bonds present in alcohol or ether groups (286.3–287.0 eV), C=O functional groups (287.5–288.1 eV), carboxyl or ester groups (289.3–290 eV), and a fifth wide shake-up satellite peak representing the π–π* transitions of aromatic rings (291.2–292.1 eV) [46,50].

Concerning O1s, the curve was fitted considering four contributions corresponding to carbonyl groups (531.1–531.8 eV), epoxide and hydroxyl groups (532.3–533.3 eV), carboxylic groups (534.0–534.4 eV), and chemisorbed H_2_O or oxygen (535.5–536.1 eV) [43,46,50,51,52]. The spectra analysis revealed that surface oxygen content varies from 3.6–4.1 at % to 6.8–7.0 at % for N-doped and non-doped samples, respectively. The relatively small amount of oxygen with respect to that of carbon may be attributed to the process temperature. It is known that higher temperatures during the thermal treatment produce a decrease of the amount of oxygen functional groups [25]. The oxygen/carbon atomic ratios also confirm this, as can be seen in Table 2. The differences in these ratios for the various samples can be ascribed to an enhancement in oxidation degree from the original GO_m_. For samples from raw graphite with smaller particle sizes, the oxidation degree of GO is higher. After the thermal treatment, the final content of oxygen is slightly higher for samples from graphite with a smaller particle size.

The point of zero charge (PZC) was used to assess the surface chemistry of the graphene materials [9], because the surface charge of carbon materials is directed by the type and population of functional groups and the pH. The NrGO samples (Figure 5) show a PZC of 8.5–8.7, while rGO samples exhibit lower PZC values of 7.2–7.4. Therefore, the rGO samples have a practically neutral surface, while the NrGO surfaces are more basic. The incorporation of nitrogen atoms into the graphene structure means more electrons in comparison with carbon atoms and this fact favors the delocalization of p electrons in N-doped samples, leading to changes in the hydrophobicity of the surface. This excess of electrons produces a higher basic strength of NrGO surfaces [47].

Raman spectroscopy is a very useful tool to evaluate the degree of disorder in the structure of graphene [8]. Figure 6 shows Raman spectra for rGO_m_-3 and NrGO_m_-4 samples. As Raman spectra at discrete spots cannot provide an overall picture in the case of non-uniform defects distributions in the sample, spectral mapping was used to acquire 25 points over a 50 × 50 μm^2^ area. Two main peaks corresponding to vibrations with E2g symmetry in the graphitic lattice (G band) at 1580 cm^−1^ and to graphite edges or structural defects (D bands) at 1345 cm^−1^ were observed. Another two featured peaks have been reported in previous studies. A band D′ peak appears at 1625 cm^−1^ as a shoulder of the G band. It arises from alterations in the tension of sp^2^ carbon atoms in the lattice caused by the arrangement of the electronic cloud [50]. A second peak assigned as 2D (historically called G’) is always present at 2700 cm^−1^ in the spectra of graphene materials [40].

The intensity ratio of the D and G bands (I_D_/I_G_) can be used as a quantitative indicator of the amount of disorder or edges within the structure of the samples. I_D_/I_G_ ratios were calculated (Table 2). A comparison of rGO_m_ samples with their NrGO_m_ counterparts shows that the ratio increases when N is introduced in the reduced graphite oxide. In addition, an increase with N content in the NrGO_m_ series is observed. This is due to a disruption of the symmetry of the lattice produced by the incorporation of heteroatoms into the graphitic structure. This effect has been described previously by Chen et al. [53]. They claimed that the introduction of N into the carbon lattice could produce distortions, transforming the graphitic region into an sp^3^ domain. From the Raman spectra, it can be observed that an increment in the N content leads to a shift of the D band to lower frequencies.

For a bulk graphite sample, the 2D band consists of two contributions. For single-layer graphene, the 2D band appears as a single sharp peak at the lower frequency (around 2690 cm^−1^). As the number of layers increases, the 2D band changes its shape, width, and position, and the G peak position shifts to lower frequencies [8]. A systematic study of the in-plane crystallite size was carried out in 1970 by Tuinstra and Koenig [54]. They found that the ratio of the D and G band intensities (I_D_/I_G_) is inversely proportional to the in-plane crystallite sizes (L_a_). The crystallite sizes (L_a_) can be calculated from L_a_(nm) = (2.4 × 10^−10^) λ^4^(I_D_/I_G_)^−1^ (λ being the Raman excitation wavelength) [55]. L_a_ were 25.6–33.1 nm for NrGO samples and 30.5–49.3 nm for rGO samples; it is concluded that the crystallite size decreases with the presence of defects, and therefore the doping level. It is consistent with the bibliography [40], which point outs that since L_a_ is the average interdefect distance, the introduction of nitrogen atoms accompanied by defects implies a smaller L_a_.

The stability of the prepared graphenic materials in air atmosphere with the temperature was studied by thermal gravimetric analysis (Appendix A, ESI). Differential thermogravimetric profiles for rGO_m_-3 samples show pronounced peaks near 598–642 °C that could be attributed to a weight loss due to the oxidation of a graphitized carbon structure. In comparison, for NrGO_m_-4 samples, this peak moved toward higher temperatures, showing oxidation temperatures between 663 and 671 °C. These findings reveal that the presence of nitrogen in the NrGO samples rise the stability in air of the graphenic material, which is in agreement with previous results [33].

### 3.2. ORR Electrocatalytic Activities

To assess the electrocatalyst’s properties toward ORR of the N-doped (NrGO_325_-4, NrGO_100_-4, and NrGO_10_-4) and non-doped (rGO_325_-3, rGO_100_-3, and rGO_10_-3) graphenic materials, a cyclic voltammetry study was initially performed in a KOH electrolyte (0.1 mol dm^−3^) saturated with N_2_ and O_2_. Appendix A (ESI) shows the CVs in the N_2_ and O_2_ electrolytes for the six samples tested. In the presence of N_2_, no redox processes are detected, while in the electrolyte purged with O_2_, the six graphene materials studied showed an irreversible ORR peak with 0.74 ≥ *E*_pc_ ≥ 0.72 V versus RHE. For comparison, the commercial Pt/C (20 wt %) was also measured in the same experimental conditions toward ORR (Appendix A, ESI), presenting analogous behavior to the graphene materials with *E*_pc_ = 0.88 V versus RHE. 

Then, the electrocatalytic properties toward ORR were further studied by LSV with an RDE at several rotation speeds in KOH purged with O_2_. The LSVs of all graphene materials are presented in Figure 7, while those corresponding to Pt/C can be seen in Appendix A. All graphene materials presented a linear relationship between *j*_L_ and rotation speed, suggesting that the electron transfer reaction is diffusion limited. The LSVs show three different regions: for *E* higher than ≈ 0.80 V, the process is kinetically controlled; potential values between ≈ 0.80 V and ≈ 0.50 V indicate the mixed kinetic-diffusion region, and for *E* ≤ 0.50 V, the process is controlled by O_2_ diffusion.

Table 3 shows comparisons of the obtained *E*_onset_ and *j*_L, 0.26V, 1600rpm_ values for the N-doped and non-doped graphene materials at 1600 rpm. All the ECs showed similar onset potentials (0.82 ≥ *E*_onset_ ≥ 0.79 V versus RHE), but higher *j_L_*_, 0.26V, 1600rpm_ values were obtained for the N-doped materials. For the non-doped graphene materials, these values varied between −2.92 and −3.46 mA cm^−2^, while for the N-doped materials, they varied between −3.03 and −4.05 mA cm^−2^. The NrGO_325_-4 electrocatalyst showed the most promising result of all the prepared graphenic materials with the more positive *E*_onset_ of 0.82 V versus RHE (0.12 V more negative than Pt/C) and the higher *j*_L, 0.26V, 1600rpm_ of −4.05 mA cm^−2^. The introduction of nitrogen species led to an improvement in the ORR properties, which is in agreement with several works that have reported the enhancement of ORR electrocatalytic activity after the N-doping of carbon materials [56,57]. Furthermore, comparing the results for the non-doped graphenic materials, it seems that the particle size of the starting graphite does not have a clear direct relationship with their ORR properties, unlike is observed for the N-doped materials. For the latter, as the particle size of the starting graphite decreased, there was an improvement of the ORR features (more positive *E*_onset_ values and higher *j*_L_), and this is a consequence of the higher degree of oxidation that favored the incorporation of increased amounts of nitrogen, leading to higher possible active sites for oxygen reduction. This is supported by the elemental analysis results where the percentages of nitrogen obtained were 5.0%, 3.8%, and 1.8% for NrGO_325_-4, NrGO_100_-4, and NrGO_10_-4, respectively.

Then, the ORR kinetics at different potentials were assessed with the Koutecky-Levich (K-L) plots obtained with the application of the K-L equation to the LSVs (in Figure 7) at rotation speeds in the range of 400 to 3000 rpm. The K-L plots of the graphene materials are presented in Appendix A, and those corresponding to Pt/C are shown in Appendix A. For the graphene materials, the *j*^−1^ increases with increasing *ω*^−1/2^, which suggests a first-order electrocatalytic oxygen reduction with respect to the concentration of oxygen dissolved. Additionally, all the K-L plots of the non-doped materials show different slopes, suggesting that *ñ**_O2_* depends on the applied potential, while for the N-doped ones, the differences are less significant. This suggests that in this case, n_O2_ is less dependent on the potential. The ORR process in alkaline medium can proceed by two pathways: a direct one involving one step (Equation 2) or an indirect one involving two steps (Equation 3 and 4) [reference 2 in ESI].
O_2_ + H_2_O + 4e^−^ → HO^−^(2)
O_2_ + H_2_O + 2e^−^ → HO_2_^−^ + HO^−^(3)
HO_2_^−^ + H_2_O + 2e^−^ → 3HO^−^(4)

The mean *n_O2_* (*ñ*_O2_) values estimated from Equation 3 in ESI are depicted in Table 3, and Figure 8a shows the changes of *n*_O2_ with the applied potential. For the non-doped materials, as the potential decreases from 0.46 to 0.26 V versus RHE, there is an increase in n_O2_ values with ΔE values of 1.0, 0.53, and 0.71 V for rGO_10_-3, rGO_100_-3, and rGO_325_-3, respectively. The variation for the N-doped materials is less significant with 0.40 ≥ ΔE ≥ 0.22 V versus RHE. The results of mean n_O2_ values obtained suggest that rGO_100_-3 and rGO_325_-3 proceed through the indirect pathway, while rGO_10_-3, NrGO_10_-4, and NrGO_100_-4 seem to proceed through a mixed mechanism, similar to *ñ*_O2_ ≈ 3. This behavior has already been reported for other N-doped structures [58]. On the other hand, for the NrGO_325_-4, a one-step four-electron transfer mechanism seems to be the leading process, since *ñ_O2_* = 3.9. This value is equal to the one obtained for Pt/C. Additionally, this result is better than that obtained for the other N-doped carbon nanotubes and graphene [32,59,60].

The better ORR performances (*j*_L_ and *ñ*_O2_) of the N-doped materials in comparison with the non-doped may be related to the presence of nitrogen as explained before, while the differences between the three N-doped materials to the percentages and types of N species found could also be related to the particle size of the starting graphite. As discussed above in Section 3.1, usually a higher degree of oxidation is obtained in GO samples with smaller graphite particle size, and on the other hand, oxygen functional groups are responsible for reacting with NH_3_, leading to the incorporation of nitrogen in the structure, which will favor ORR. Elemental analysis showed that NrGO_325_-4, NrGO_100_-4, and NrGO_10_-4 presented 5.0%, 3.8%, and 1.8% of N, respectively, whereas the XPS results indicated percentages of 3.4%, 3.3%, and 3.2%. The NrGO_325_-4 sample presents a higher N% at the surface and has also a higher percentage of pyridinic nitrogen. The latest have been reported to work as ORR electrocatalytic sites due to the electron donated to the conjugated π-bond of graphene and to the lone pair of electrons they have [61,62].

To confirm the excellent result obtained for NrGO_325_-4, RRDE tests were performed. The *j* values obtained at the disk and ring are depicted in Appendix A. For comparison, the Pt/C data were also included. Equation (1) was used to estimate the percentage of H_2_O_2_ produced, and the results are presented in Appendix A at several potential values. The NrGO_325_-4 electrocatalyst presented a low percentage of H_2_O_2_ (≈ 15%), which is in accordance with the high *ñ*_O2_ value (*ñ*_O2_ = 3.9), while a value of 3% was obtained for Pt/C.

Tafel plots (Figure 8b) were obtained from LSV data in Appendix A in KOH electrolyte purged with O_2_ at 1600 rpm. The Tafel slopes obtained between *E* = 1.00 to 0.70 V versus RHE were 65, 48, 98, 78, 99, 62, and 89 mv dec^−1^ for rGO_10_-3, rGO_100_-3, rGO_325_-3, NrGO_10_-4, NrGO_100_-4, NrGO_325_-4, and Pt/C, respectively. All electrocatalysts except for rGO_325_-3 and NrGO_100_-4 presented Tafel slopes lower than those obtained for Pt/C, which suggests that the graphene materials can easily adsorb O_2_ onto their surface and activate it, promoting a robust electrocatalytic activity toward ORR. The different Tafel slopes obtained for rGO_325_-3 and NrGO_100_-4 are most likely associated to fluctuations in the oxygenated intermediates adsorption strength or to an incomplete electrocatalyst, which is used as a consequence of mass transport losses [63,64]. It is recognized that the intermediates adsorption strength is dependent on the physical properties and on the chemical nature of the selected electrocatalyst, which rules the determining step rate. For these two electrocatalysts, the first discharge step upon the consumption of the MOOH with a high coverage of MOO^−^ is the rate-determining step, whereas for the remaining electrocatalysts, the conversion of MOO^−^ to MOOH rules the overall reaction rate (where M stands for an empty site on the electrocatalyst surface) [63].

Tolerance to methanol crossover is another parameter that is usually evaluated because in fuels cells run on methanol, this can be a problem, as the performance of the cathode can be drastically reduced if the catalyst in sensitive to methanol. Therefore, to evaluate this parameter, chronoamperometric tests were performed in a KOH electrolyte purged with oxygen for 2500 s, to which 0.5 mol dm^−3^ of methanol were added at *t* = 500 s (Figure 8c). The presence of methanol led to an impressive decrease (≈ 49%) of Pt/C current. Oppositely, the graphene materials are much more stable and less sensible to methanol with current retentions between 78% and 92% suggesting higher selectivity toward oxygen reduction.

The EC stability is also of extreme importance. So, this was evaluated, for all graphene materials, using chronoamperometric runs in KOH electrolyte (in O_2_) for 20,000 s applying a potential of *E* = 0.50 V versus RHE at 1600 rpm. The results obtained for Pt/C and all graphenic materials are presented in Figure 8d. The best result was observed for Pt/C with a current retention of 88%. The NrGO_325_-4 electrocatalyst showed the best result from all the graphene materials with a current retention of 85%, which is only 3% less than Pt/C. For the other ECs, the current retention percentages were 63%, 72%, 82%, 80%, and 78% for rGO_10_-3, rGO_100_-3, rGO_325_-3, NrGO_10_-4, and NrGO_100_-4, respectively. The best performance of the NrGO_325_-4 electrocatalyst can be ascribed to the combination of two features, the N-doping and the smaller particle size of the starting graphite.

## 4. Conclusions

In this paper, we report a viable method for the synthesis of graphene materials and nitrogen-doped reduced graphene oxide derivatives with enhanced properties. These are based on the chemical oxidation of graphite flakes with different particle sizes and selecting the experimental conditions used during the subsequent thermal exfoliation process. 

The characterization results suggest that the starting particle size and thermal conditions applied during the exfoliation treatment remarkably affect the final surface properties of the prepared materials. The results point out that smaller particle sizes lead to higher surface areas. We achieved surface areas of 867 m^2^g^−1^ for rGO. For NrGO samples obtained from graphite 10, 100, and 325, we observed BET surface areas of 236, 420, and 492 m^2^g^−1^ respectively, which may provide a new way to control the surface area of N-doped graphene.

The use of different reduction atmosphere (NH_3_ versus inert) allow to successfully introduce nitrogen within the graphenic structure, enhancing the electronic properties and basicity of the doped materials. For the N-doped samples, the amount of nitrogen introduced and surface areas could be also tailored. 

As a result of their tailored enhanced properties, these optimized NrGO and rGO samples were successfully applied as ORR electrocatalysts. The NrGO samples showed better ORR performance in alkaline medium with onset potentials ranging from 0.79 and 0.82 V versus RHE, low Tafel slopes (62–99 mV dec^−1^), and good *j*_L_ values (−3.03–−4.05 mA cm^−2^). The better performance of NrGO_325_-4 was attributed not only to the higher content of nitrogen but also to the smaller particle size of the starting material. Moreover, all the graphene materials presented good durability/stability and very low sensitivity to methanol. This work has led to a new class of metal-free ORR electrocatalysts with good efficiency and stability.

## Figures and Tables

**Figure 1 nanomaterials-09-01761-f001:**
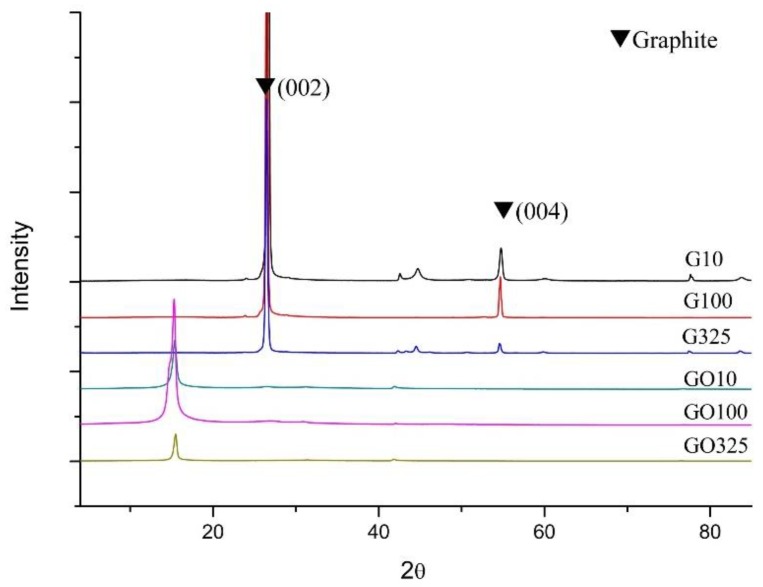
XRD patterns of graphene (G_m_) and graphite oxide (GO_m_), where m indicates the mesh samples used.

**Figure 2 nanomaterials-09-01761-f002:**
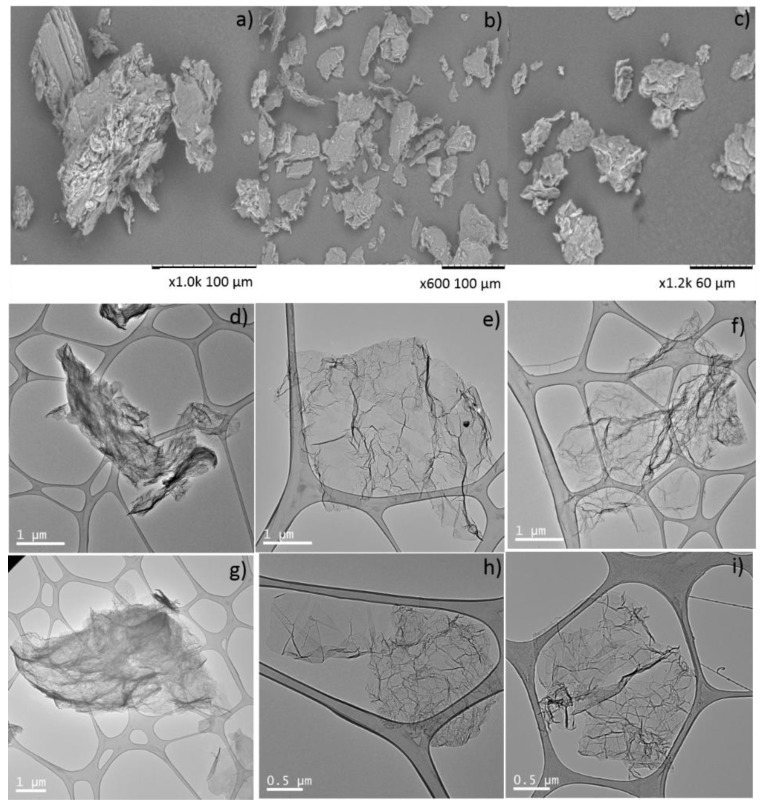
Representative SEM images of (**a**) GO_10_, (**b**) GO_100_ and (**c**) GO_325_ samples. TEM micrographs for (**d**) rGO_10_-3, (**e**) rGO_100_-3, (**f**) rGO_325_-3, (**g**) NrGO_10_-4, (**h**) NrGO_100_-4 and (**i**) NrGO_325_-4 samples. NrGO: N-doped reduced graphene oxide, rGO: reduced graphene oxide.

**Figure 3 nanomaterials-09-01761-f003:**
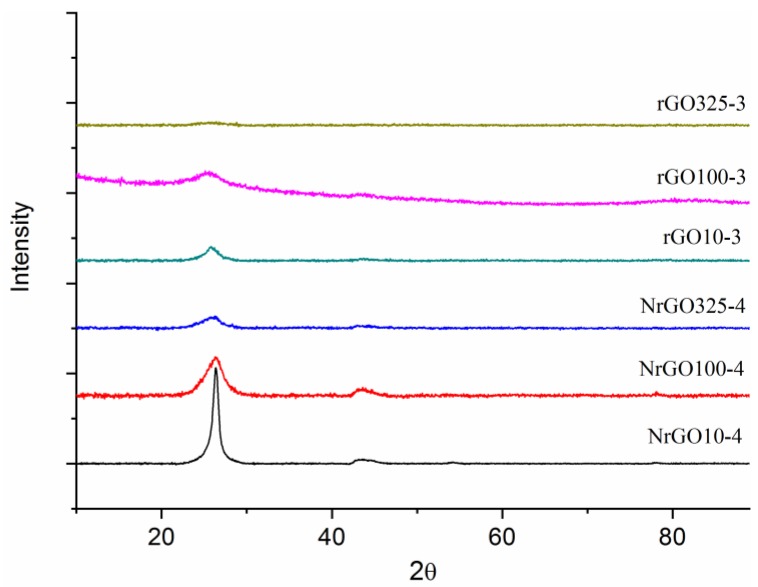
XRD patterns of reduced graphene materials.

**Figure 4 nanomaterials-09-01761-f004:**
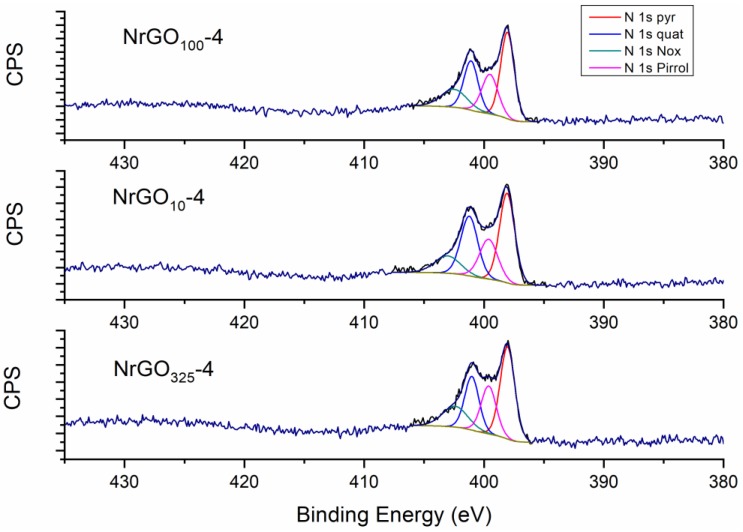
XPS spectra of the N 1s region for NrGO100-4, NrGO10-4, and NrGO325-4 samples.

**Figure 5 nanomaterials-09-01761-f005:**
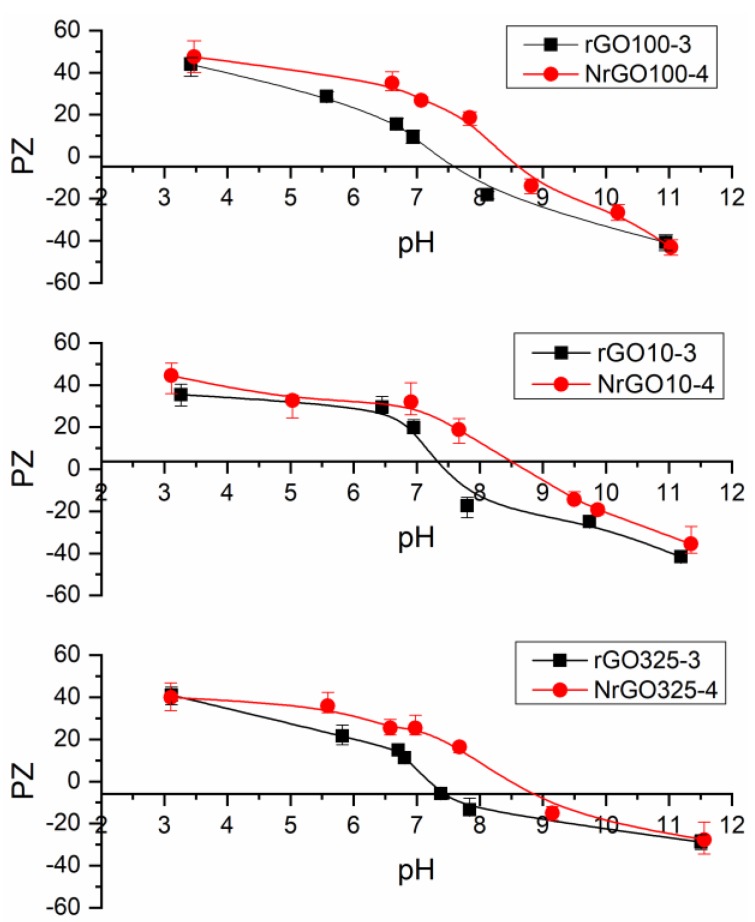
PZC for rGOm-3 and N-doped reduced graphene oxide (NrGOm-4) samples.

**Figure 6 nanomaterials-09-01761-f006:**
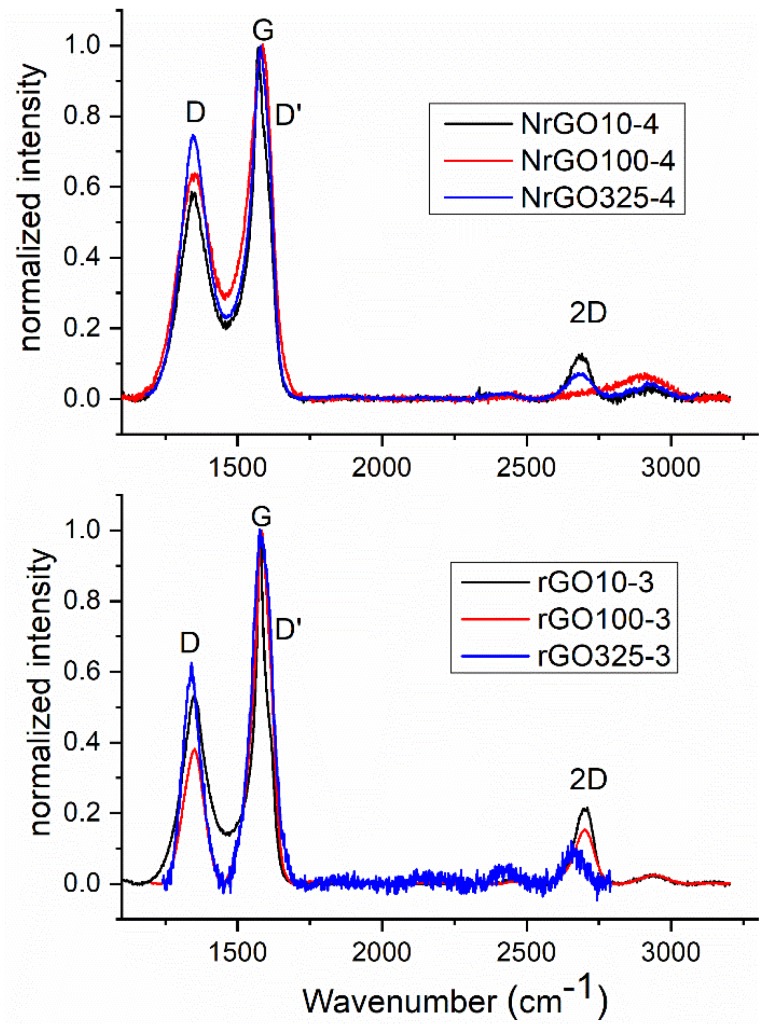
Raman spectra for rGO_m_-3 and NrGO_m_-4 samples.

**Figure 7 nanomaterials-09-01761-f007:**
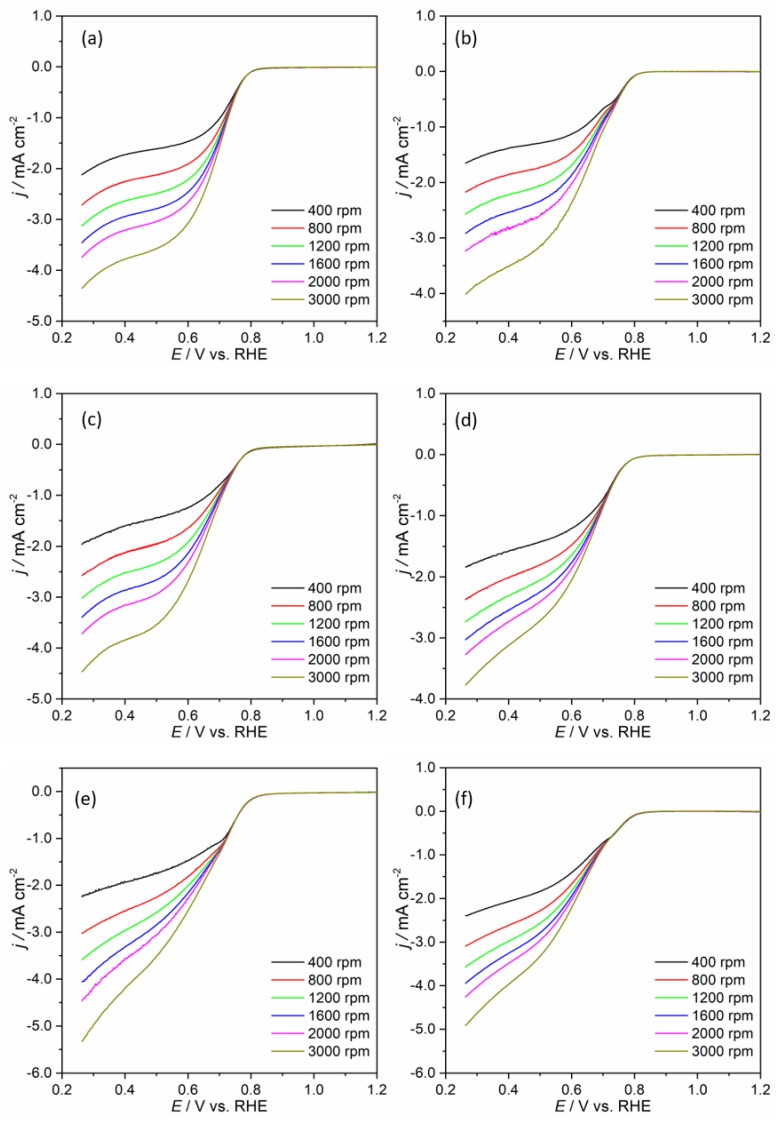
Oxygen reduction reaction (ORR) polarization curves of rGO_10_-3 (**a**), rGO_100_-3 (**b**), rGO_325_-3 (**c**), NrGO_10_-4 (**d**), NrGO_100_-4 (**e**), and NrGO_325_-4 (**f**) modified electrodes at different rotation rates in KOH purged with O_2_ at 0.005 V s^−1^.

**Figure 8 nanomaterials-09-01761-f008:**
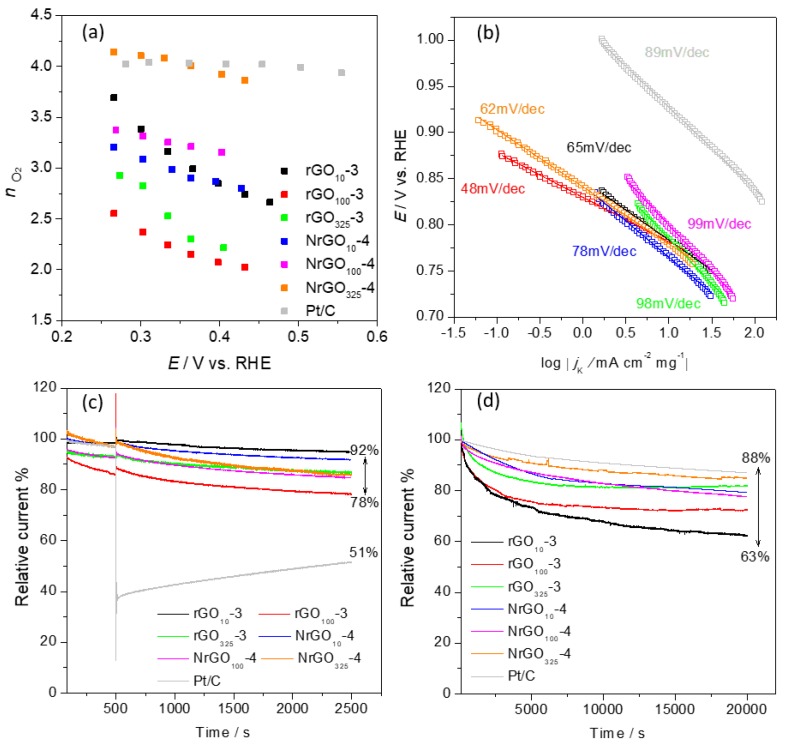
*n_O2_* at several potential values (**a**); ORR Tafel plots (**b**); chronoamperograms in O_2_-saturated KOH electrolyte at 1600 rpm and *E* = 0.50 V vs. reversible hydrogen electrode (RHE), with the addition of methanol (0.5 mol dm^−3^) at *t* = 500 s (**c**) and without for 20000 s (**d**).

**Table 1 nanomaterials-09-01761-t001:** Interlayer distance d_(002)_, estimated number of layers (N_L_), S_BET_, and N content (%) for N doped and non-doped reduced graphene oxide samples.

GO_m_	Ramp	Atmosphere	Sample	d_(002)_ (nm)	N_L_*	S_BET_ (m^2^ g^−1^)	N (%)
GO_325_	1	Inert	rGO_325_-1	0.35	13	767	-
	2		rGO_325_-2	0.35	10	804	-
	3		rGO_325_-3	0.34	12	867	-
	4		rGO_325_-4	0.35	17	667	-
	5		rGO_325_-5	0.36	26	866	-
GO_100_	3	Inert	rGO_100_-3	0.35	10	778	-
GO_10_	3	Inert	rGO_10_-3	0.34	18	505	-
GO_325_	1	Ammonia	NrGO_325_-1	0.34	14	428	4.8
	2		NrGO_325_-2	0.35	9	427	4.4
	3		NrGO_325_-3	0.34	14	460	5.0
	4		NrGO_325_-4	0.34	14	492	5.0
	5		NrGO_325_-5	0.34	13	476	4.1
GO_100_	4	Ammonia	NrGO_100_-4	0.35	10	420	3.8
GO_10_	4	Ammonia	NrGO_10_-4	0.34	40	236	1.8

* N_L_= (L_002_ + d_002_)/d_002_.

**Table 2 nanomaterials-09-01761-t002:** XPS deconvolution results and RAMAN I_D_/I_G_ ratio for rGO_m_-3 and NrGO_m_-4 samples.

Sample	O (at %)	N (at %)	O 1s				N1s				I_D_/I_G_
			C–O	C=O	COOH	H_2_O	Pyr	Pyrr	Quat	Nox	
rGO_325_-3	7.0	-	57.0	21.1	15.7	6.2	-	-	-	-	0.63
rGO_100_-3	7.0	-	63.1	17.0	13.7	6.3	-	-	-	-	0.39
rGO_10_-3	6.8	-	54.6	20.7	18.7	6.0	-	-	-	-	0.53
NrGO_325_-4	4.1	3.4	54.7	23.5	17.6	4.2	40.0	21.7	22.7	15.6	0.75
NrGO_100_-4	3.6	3.3	54.2	21.2	18.9	5.7	39.5	21.7	23.6	15.3	0.63
NrGO_10_-4	3.6	3.2	56.2	17.4	18.7	7.6	38.5	20.5	28.5	12.4	0.58

**Table 3 nanomaterials-09-01761-t003:** Relevant ORR parameters for commercial Pt/C and activated carbons prepared.

Sample	*E*_onset_ vs. RHE	*j*_L_ (mA cm^−2^)	*ñ* _O2_	Tafel Slopes (mV dec^−1^) ^*^
Pt/C (20 wt %)	0.94	−4.70	4.0	89
rGO_325_-3	0.81	−3.40	2.6	98
rGO_100_-3	0.80	−2.92	2.2	48
rGO_10_-3	0.80	−3.46	3.1	65
NrGO_325_-4	0.82	−4.05	3.9	62
NrGO_100_-4	0.80	−3.94	3.3	99
NrGO_10_-4	0.79	−3.03	3.0	78

* normalized by the mass of catalysts

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
