# Peer review of "Upgrading the Properties of Reduced Graphene Oxide and Nitrogen-Doped Reduced Graphene Oxide Produced by Thermal Reduction toward Efficient ORR Electrocatalysts"

_nanomaterials, 2019, doi:10.3390/nano9121761_

Round 1

Reviewer 1 Report

This contribution from the group of Rodríguez-Ramos describes the N-doped (NrGO) and non-doped (rGO) graphenic materials by oxidationand further thermal treatment under ammonia and inert atmospheres, respectively, of natural graphites of different particle sizes. The author reports a serics of graphenic materials with differert condictions. for examples, selecting adequate graphite sizes and convenient experimental conditions ...etc. The authors accurately explain how the data were collected. There is sufficient information that the experiment can be reproduced. I believe that the work can be published in its current form.

I recommend the paper is accepted with some minor changes.

Minor point:

Fig 4, 6 and 8 are not very clear.

The work discussedfive different exfoliation ramps in the experimental section. However, the N contents are similar in table 1 (4.1 to 4.8%). Why? And in Fig 2 TEM images, what different between rGo-x and NrGo-x? Authors need to describe the differences.

In conclusions, line 488, “The better performance of NrGO325-4 was attributed not only to *the higher content of nitrogen* but also to the smaller *particle size* of the starting material.” Which one is the most important condiction for the OOR electrocatalyst? I think it can be clarified.

Page 7, line 207. SBET is wrong.

as stated above, after the author changes those points, this paper can be accepted in this journal.

Reviewer 2 Report

The results pointing out possibility to be able to tailor various properties of rGO and NrGO are very useful for readers. The discussions for the experimental results are also careful.

Reviewer 3 Report

Dear Editors,

It was a pleasure for me to read the paper entitled by "Upgrading the properties of reduced graphene oxide 2 and nitrogen doped reduced graphene oxide 3 produced by thermal reduction towards efficient 4 ORR electrocatalysts" by Carolina S. Ramirez-Barria. This solid work is devoted to synthesis and comprehensive characterization of the selected graphenic materials with tunable and well controlled propererties. Authors showed that the obtained graphenic materials are very promising as efficient ORR electrocatalysts. All conclusions are well supported by results of XPS, XRD, Raman spectroscopy, SEM and electrochemical measurements. The paper is well organized, and authors give reasonable explanations for all observed effects. This work looks very interesting for broad readers' audience, especially for graphene community. I am glad to recommend this paper for publishing in Nanomaterials after improvement of English.

Round 2

Reviewer 1 Report

The revised draft is an improvement on that submitted initially and is now acceptable for publication